# Immune Responses to HBV Vaccine in People Living with HIV (PLWHs) Who Achieved Successful Treatment: A Prospective Cohort Study

**DOI:** 10.3390/vaccines11020400

**Published:** 2023-02-09

**Authors:** Ling Xu, Li Zhang, Shuang Kang, Xiaodi Li, Lianfeng Lu, Xiaosheng Liu, Xiaojing Song, Yanling Li, Xiaoxia Li, Wei Lyu, Wei Cao, Zhengyin Liu, Taisheng Li

**Affiliations:** 1Department of Infectious Diseases, Peking Union Medical College Hospital, Chinese Academy of Medical Sciences & Peking Union Medical College, Beijing 100730, China; 2Center for AIDS Research, Chinese Academy of Medical Sciences & Peking Union Medical College, Beijing 100730, China; 3Department of Infectious Diseases and Clinical Microbiology, Beijing Chao-yang Hospital, Capital Medical University, Beijing 100020, China; 4School of Medicine, Tsinghua University, Beijing 100084, China

**Keywords:** PLWH, HBV vaccine, total HIV DNA levels, IFN-r, TNF-a

## Abstract

Background: Understanding immune responses after HBV vaccination is important to prevent HBV infection in PLWH and to achieve successful treatment. Methods: Thirty-two PLWHs with CD4^+^ cell count > 350 cells/µL and HIV RNA < 200 copies/mL were vaccinated with 20 µg of HBV vaccine at weeks 0, 4, and 24 in this prospective study. We measured total HIV DNA levels, HBsAb titers and HBsAg-specific T-cell responses during follow-up time. Results: All patients achieved protective HBsAb titer after immunization. The magnitude of the IFN-r and TNF-a response to HBsAg was 22.0 (IQR: 6.5–65.0) and 106.50 (IQR: 58.5–203.0) spot-forming cells (SFC)/10^5^ PBMC, respectively at week 0. The level of IFN-r secreted at weeks 12 and weeks 36 to 48 was comparable with that at week 0. However, IFN-r response was higher at weeks 12 than that at weeks 36 to 48 (*p* = 0.02). The level of TNF-a secreted at weeks 12 was higher than that at week 0 (*p* < 0.001). Total HIV DNA levels were 2.76 (IQR: 2.47–3.07), 2.77 (IQR: 2.50–3.09), 2.77(IQR: 2.41–2.89) log_10_ copies/10^6^ PBMCs at weeks 0, 12, 36 to 48, respectively. No correlation was observed between IFN-r and TNF-a levels and HBsAb titer as well as total HIV DNA levels after immunization. Conclusion: Humoral immunity was satisfactory, but cellular immunity and decline in HIV reservoir were not optimal after HBV vaccine immunization in these patients.

## 1. Introduction

Approximately 7.5% of people living with human immunodeficiency virus (PLWHs) worldwide are coinfected with hepatitis B virus (HBV) due to the same transmission routes [1]. The coinfection prevalence varies in different regions. For example, it is higher than 10% in sub-Saharan Africa, west Africa, east Asia, and India [1,2]. HIV/HBV-coinfected patients are more likely to develop liver-associated diseases [3,4].

Some patients treated with a TDF or TAF-based regimens are protected from HBV infection [5]. However, the new long-acting drugs such as cabotegravir and rilpivirine cannot protect from HBV, and people with HBV are not eligible for this treatment [6]. Therefore, vaccination is strongly suggested for a better treatment choice and, consequently, a better quality of life in PLWHs with negative hepatitis B serology [7]. The standard recombinant HBV vaccine could achieve 95% seroprotection in HIV-negative adults [8,9]. Unfortunately, the immunogenicity of the HBV vaccine achieved in PLWH ranges from 35% to 70% [10]. Furthermore, a study demonstrated that PLWHs who reached optimal vaccine responses failed to maintain HBV immunity for a long time [11]. The mean time to lose an effective antibody to hepatitis B surface antigen (HBsAb) was 2, 3.7, and 4.4 years for patients with an HBsAb titer of 10–100 IU/I, >100–1000 IU/I, and >1000 IU/I at primary vaccination, respectively [12]. Age, dose of the HBV vaccine, CD4^+^ cell counts, low HIV RNA levels and combination antiretroviral therapy (cART) were associated with HBV vaccine responses in PLWHs [10,13,14].

Cellular immunity plays a crucial role in HBV vaccination. Previous studies have evaluated cellular immune responses to HBV vaccination by different variables in HIV-negative people [15,16]. Simons et al. demonstrated that interferon-gamma (IFN-r) and tumor necrosis factor-alpha (TNF-a)-based T-cell response to hepatitis B surface antigen (HBsAg) could last at least 32 years regardless of HBsAb level in HIV-negative subjects vaccinated with HBV vaccine during childhood [16]. However, Chawansuntati et al. found no statistical differences in the frequencies of IFN-r and TNF-a production of total or memory CD4^+^ or CD8^+^ cells during the 12 months of study in PLWHs vaccinated with standard or double doses of HBV vaccine at 0, 1, 6 months [17].

CD4^+^ cells play an important role in the process of HBV antibody production following vaccination. Then, effective vaccination could produce long-lived memory T cells that recognize and respond to HBV infection. However, central memory CD4^+^ cells contribute to maintain the viral reservoir in PLWHs [18]. Bekele et al. demonstrated the influence of HBV vaccination in affecting the size of HIV reservoirs in 22 HIV-infected children. They showed that total HIV DNA levels after both hepatitis A virus (HAV) and HBV vaccination were reduced compared to baseline, although the decline was not statistically significant [19]. The influence of HBV vaccination in the size of the HIV reservoir in PLWHs has not been well evaluated.

CD4^+^ cell count > 350 cells/µL and HIV RNA < 200 copies/mL after cART are considered as a successful treatment in PLWHs [20]. The humoral and cellular immune responses to HBV vaccination in these satisfactory patients when they are vaccinated with 20 µg of recombinant HBV DNA vaccine at weeks 0, 4, and 24 are unknown at present. Could HBV vaccination reduce the size of HIV reservoirs in HIV-infected adults? Based on these facts, we aim to observe the long-term humoral immunity to HBV vaccination and to analyze the changes in cellular immunity as well as HIV reservoirs during the follow-up period in a prospective study.

## 2. Materials and Methods

### 2.1. Study Design and Population

We performed a prospective study from December 2018 to January 2020 at the Department of Infectious Disease at Peking Union Medical College Hospital (PUMCH) clinics. We included 32 PLWHs over 18 years old. They were seronegative for HBsAg, HBsAb, antibody to hepatitis B core antigen (anti-HBc), and antibody to hepatitis C virus (anti-HCV), and without a history of HBV vaccination in previous five years. Patients were eligible to participate if they received cART and had CD4^+^ cell count >350 cells/µL and HIV viral load <200 copies/mL for the past 6 months. Exclusion criteria included being pregnant or breastfeeding; acute elevations of liver enzymes within the past three months (alanine aminotransferase (ALT) or aspartate aminotransferase (AST) two or more times the normal upper); having a history of hypersensitivity to any component of the vaccine or other immunocompromised conditions Furthermore HIV.

The enrolled individuals were given 20 ug recombinant HBV DNA vaccine at weeks 0, 4, and 24. The diagram of the participants is shown in Appendix A. Ten participants did not vaccinate for personal reasons. The main reason was that they needed to take three weeks of leave to vaccinate, which they thought would affect their work. Furthermore, they still had concerns about the safety and effectiveness of vaccines. The patients were followed-up every 12 weeks in our clinics; therefore, HBsAb was tested, and full blood was collected to separate PBMC at weeks 0, 4, and 36 in the original plan. However, some patients could not come for follow-up at week 36 due to the outbreak of the COVID-19 epidemic. Thus, they were followed-up between weeks 36 and 48 (Appendix A). Furthermore, HBsAb was tested annually in every patient, whether vaccinated or not, in order to evaluate HBV infection in our clinics. Therefore, we could observe the dynamics of HBsAb titer in longer follow-up duration for enrolled participants. Enzyme-linked immunosorbent assay (Elisa) was used to measure HBsAb, and the cut-off value of this test was 10 IU/mL. The serological response was defined as HBsAb > 10 IU/mL.

### 2.2. Total HIV DNA Determination

Total HIV DNA was extracted from peripheral blood containing 0.25–1 million PBMCs using the QIAsymphony DNA Mini Kit (QIAGEN, Hilden, Germany). The SUPBIO total HIV DNA Quantitative PCR Kit (SUPBIO, Guangzhou, China) was used for simultaneously quantitating total HIV DNA and cell number, following the manufacturer’s instructions as previously described [21]. The linear quantification range of the SUPBIO total HIV DNA quantitative kit was 20 copies/10^6^ PBMCs to 100,000 copies/10^6^ PBMCs.

### 2.3. Lymphocyte Subsets Phenotyping

Immunophenotyping of peripheral blood lymphocytes was analyzed by three-color flow cytometry (Epics XL flow cytometry; Bechman Coulter, USA) using commercially available monoclonal antibodies as previously described [22]. The percentages and counts of the following lymphocyte subsets were measured, including CD3^−^CD16^+^ CD56^+^ NK cells, CD3^−^CD4^+^ cells, CD3^−^CD8^+^ cells, memory CD4^+^ cells, CD4^+^ CD45RA^+^CD62^+^ naïve cells, CD8^+^ HLA-DR^+^ cells, CD8^+^ CD38^+^ cells, and the CD4/CD8 ratio. The memory CD4^+^ cells were counted by difference between CD4^+^ cells and naïve CD4^+^ cells. Cell counts of lymphocyte subsets were calculated using a dual-platform method with white blood cell counts and lymphocyte differentials obtained from routine blood tests of the same specimen.

### 2.4. Human IFN-r and TNF-a ELISpot Assay

Isolated PBMCs were incubated at 37 degrees centigrade in 5% CO_2_ overnight. PBMCs (1.0 × 10^5^) were pulsed with 1 ug/mL recombinant hepatitis B surface protein for 24 h at 37 degrees centigrade in 5% CO_2_, and then, we tested the cellular immunity responses using a standard Human IFN-r and TNF-a enzyme-linked immunosorbent spot (ELISpot) assay in triplicate wells. Wells containing PBMCs with and without anti-CD3 were used as positive and negative controls, respectively. To quantify antigen-specific responses, mean spots of the control wells were subtracted from the positive wells, and the results were expressed as spot-forming cells (SFC) per 10^5^ PBMCs. Responses were considered positive if results were at least twice the mean of the triplicate negative control wells and >30 SFCs/10^5^ PBMCs.

### 2.5. Ethics Statement

The Institutional Review Board of Peking Union Medical College Hospital (PUMCH) approved the parent studies, and each participant provided written informed consent.

### 2.6. Statistical Analysis

Analyses were performed using SPSS 23.0 (IBM Corp, Armonk, NY, USA), with *p* values < 0.05 signifying statistical significance. Descriptive statistics were presented as median (M) with interquartile ranges (IQRS). The Mann–Whitney U test was conducted for comparison of noncategorical variables. Categorical variables were analyzed by Chi-squared test or Fisher exact test. Association between continuous variables was tested using a nonparametric Spearman rank correlation test.

## 3. Results

### 3.1. Characteristics of the Patients

From a total of 1032 follow-up patients seeking HIV care in PUMCH, 61 were eligible for this study. Reasons for exclusion are detailed in Appendix A. Out of the 32 patients who initiated vaccination, the median age was 36 (IQR: 30–50) years, and 29 (90.6%) were male. In total, 75.0% of individuals were infected with HIV by homosexual transmission route. The median nadir CD4^+^ cell count was 234 (IQR: 91–350) cells/µL, CD4/CD8 ratio was 0.32 (IQR: 0.19–0.38) and HIV RNA load was 4.85 (IQR:4.33–5.30) log_10_ copies/mL before cART. The median CD4^+^ cell count was 554 (IQR: 436–621) cells/µL, and HIV RNA level was suppressed in all except for two, in whom it was 68, 79 copies/mL at the first dose of their immunization (week 0). There were no relevant alterations in laboratory values, including liver function tests, hematological counts, and serum creatinine (Table 1).

### 3.2. The Dynamics of Lymphocyte Subsets and HIV RNA Levels after Receiving HBV Vaccination

At weeks 12 and 36 to 48, the CD4^+^ cell counts, including naïve CD4^+^ and memory CD4^+^ cell counts in all patients, were comparable with that at week 0. Additionally, the percentages of CD8^+^CD38^+^ (37.1 (IQR: 29.5–43.2) and 32.8 (IQR: 26.3–41.95), respectively), CD8^+^HLA-DR^+^ (37.3 (IQR: 27.0–49.8) and 41.2 (IQR: 28.7–57.0), respectively) (Appendix A), and B cell counts (171 (IQR: 89–259) cells/µL and 178 (IQR: 141–244) cells/µL, respectively) (Appendix A), were similar to that at week 0. Twenty-nine patients achieved HIV RNA load that was effectively suppressed. The remaining three patients had detectable HIV RNA levels, 55, 57, 61 copies/mL, respectively, at weeks 36 to 48. Two of the patients had detectable HIV RNA at 79 copies/mL, 68 copies/mL at week 0.

### 3.3. HBsAb Levels at Weeks 12

In response to HBV vaccination, the HBsAb levels observed in 31 patients at weeks 12 and 25 (80.6%) were positive responders with a medium HBsAb titer of 157.87 (IQR: 18.13–802.73) IU/mL. Of patients who achieved a protective HBsAb titer, six (24.0%) had anti-HBs titer above 1000 IU/mL, with 10 (40.0%) between 100 and 1000 IU/mL. Six individuals were non-responders with HBsAb < 10 IU/mL, and these subjects were older than the responders (34 (IQR: 29–41) vs. 54 (IQR: 40–57) years, *p* = 0.012).

HBsAb titer was tested between 36 and 48 weeks for 32 patients. All achieved a protective HBV titer. The medium HBsAb titer was 663.07 (334.39–1000) IU/mL, and 12 patients (37.5%) had HBsAb levels above 1000 IU/mL, with 16 (50.0%) between 100 and 1000 IU/mL. The remaining four patients (12.5%) had HBsAb levels between 10 and 100 IU/mL (Figure 1).

### 3.4. Persistence of Seroprotection following HBV Vaccination

Two or more quantitative HBsAb levels were evaluated in 32 individuals at 102 (IQR: 93–135) weeks of follow-up duration. The medium HBsAb titer was 177.8 (66.28–345.1) IU/mL. HBsAb level was lower than 10 IU/mL in three patients at 1.11, 5.64, 6.43 IU/mL. It was 25.45, 166.76, 28.78 IU/mL after HBV vaccination. Ten patients achieved HBsAb levels between 10 and 100 IU/mL, 16 patients between 100 and 1000 IU/mL, and 3 subjects with more than 1000 IU/mL. Characteristics of participants who did not sustain a protective HBV serological immunity and those who did at 102 (IQR: 93–135) weeks follow-up duration are shown in Table 2 and Table 3, respectively.

### 3.5. The Changes in HBsAg-Specific T-Cell Responses during Follow-Up

HBsAg-specific IFN-r or TNF-a-producing T cells were measured by ELISpot analysis in 30 patients. The median magnitude of the IFN-r response to HBsAg was 22 (IQR: 6.5–65.0), 39.3 (IQR: 13.8–117.0), 15.0 (IQR: 5.5–74.0) SFC/10^5^ PBMC, respectively, at weeks 0, 12, 36 to 48. The median magnitude of the TNF-a response to HBsAg was 106.5 (IQR: 58.5–203.0), 198.0 (IQR: 106.5–330.0), 170.8 (IQR: 48.6–247.1) SFC/10^5^ PBMC, respectively, at weeks 0, 12, 36 to 48. The level of IFN-r secreted at weeks 12 and weeks 36 to 48 was comparable with that at week 0. However, IFN-r response was higher at week 12 than that at weeks 36 to 48 (*p* = 0.02). The level of TNF-a secreted at week 12 was higher than that at week 0 (*p* < 0.001) (Figure 2).

We further explored whether HBsAg-specific T cell responses corresponded with HBsAb titer. We found that there was no correlation between the magnitude of the IFN-r and TNF-a responses to HBsAg and the HBsAb titer after the immunization procedure (*p* > 0.05) (Figure 3).

### 3.6. Total HIV DNA Levels after Receiving HBV Vaccination

HIV DNA levels were measured in 32, 30, 31 patients at weeks 0, 12, 36 to 48, respectively. Total HIV DNA levels were 2.76 (IQR: 2.47–3.07) log_10_ copies/10^6^ PBMCs at week 0 in the included patients. No significant alteration was observed in the total HIV DNA levels at weeks 12 and 36 to 48 weeks (2.77 (IQR: 2.50–3.09) log_10_ copies/10^6^ PBMCs, 2.77(IQR: 2.41–2.89) log_10_ copies/10^6^ PBMCs, respectively) (*p* > 0.05). No correlation was found between the total HIV DNA levels and HBsAb levels at weeks 12 and 36 to 48 weeks (r = −0.102, *p* = 0.598; r = −0.167, *p* = 0.396, respectively). Furthermore, the total HIV DNA levels were not associated with IFN-r and TNF-a levels at weeks 0, 12, and 36 to 48 (*p* > 0.05).

## 4. Discussion

In this prospective study, we enrolled 32 PLWHs who achieved successful cART, and they all completed the three-dose vaccination schedule. All patients achieved seroprotection with HBsAb > 10 IU/mL at weeks 36 to 48, indicating that successful cART could promote PLWHs to achieve similar HBV vaccination responses with HIV-negative subjects. Most of them had protective immunity at the median 102-week (IQR: 93–135) follow-up period, suggesting that humoral immunity could persist for a relatively long time. Unfortunately, cellular immunity did not significantly increase after completion of the vaccination schedule. The magnitude of the IFN-r- and TNF-a-based T-cell responses to HBsAg were highest at 12 weeks. Furthermore, we demonstrated that HBV vaccination did not affect the total HIV DNA levels, which were rarely evaluated in previous studies.

The immunogenicity of HBV vaccination in PLWHs varies in different studies, which could be related to the heterogeneous study designs, patient characteristics, and vaccination doses [10]. Landrum et al. demonstrated that higher CD4 cell count at the time of vaccination and use of cART were associated with a higher likelihood of HBV vaccine response [23]. In our study, all patients achieved protective HBsAb between 36 and 48 weeks. Furthermore, 87.5% of patients had ideal seroprotective HBsAb levels. CD4^+^ cells play an essential role in antibody production by B cells. The optimal serological response may be attributed to the high CD4^+^ cell count and ideal viral suppression. Of course, there are other cells and processes involved in producing antibodies, such as B cells and regulatory T cells [24]. Future studies are necessary to explore the recovery of these cells and the role they play in modulating immune responses in patients with CD4^+^ cell count > 350 cells/µL.

HBV vaccination provides long-term immunity in 90% of immunocompetent individuals after three 20 ug doses of anti-HBV vaccine administered over six months. Fauci et al. demonstrated that protective antibody levels persisted in 65% of healthcare workers who had received three vaccine doses [25]. Unfortunately, PLWHs have difficulty in maintaining HBV immunity. In a small sample, Laura et al. reported 69.3% and 26.9% seroprotection after 36 and 84 months among patients with successful vaccination using a 20 mcg three-dose schedule [26]. Cruciani et al. described the persistence of protective HBsAb titers of 70.6% and 32.7% at 12 and 24 months of follow-up [27]. In our study, 90.6% of patients had protective HBsAb titers at 102 (IQR: 93–135) weeks follow-up duration even though HBsAb titers waned over time. Retrospective studies have identified multiple factors affecting long-term immune response, including maximal HBsAb level following primary vaccination, current and nadir CD4^+^ cell count, HIV viral load, and CD19 cell percentage [12,28,29]. Our data indicated that cART-experienced patients with high CD4^+^ cell counts had more potency to maintain the persistence of protective HBsAb titers. Moreover, the patients who failed to sustain protective HBsAb had low anti-HBs levels after the HBV vaccination schedule was completed, agreeing with prior studies emphasizing the importance of improving responsiveness after primary immunization.

Previous studies have identified cellular immune responses to HBV vaccination in HIV-negative adults [15,30], which were limited in PLWHs. We showed that the levels of IFN-r secreted at weeks 12 and 36–48 were not significantly higher than that at week 0, which was similar to the results in a previous study [17]. This indicated that the recovery of cellular immune responses in PLWHs was pretty tricky even though ideal humoral immunity was achieved. Future studies could try to vaccinate participants with a double-dose recombinant HBV vaccine at monthly intervals and observe the dynamics of cellular immune responses.

The existence of HIV reservoirs makes curing HIV infection pretty difficult. A previous study evaluated the effect of influenza vaccination on the HIV reservoir, indicating that vaccination may reduce HIV-associated immune activation and lead to reduced HIV replication in PLWHs receiving effective treatment [31]. Our data demonstrated that HBV vaccination did not affect the size of the HIV reservoir in HIV-infected adults. This may be attributed to the fact that total HIV DNA levels decrease significantly from 0 to 1 year of cART [18]. However, patients in our study mainly received treatment for more than 1 year, when the total HIV DNA levels were relatively stable. In general, our results provide some information for the effect of HBV vaccination on the HIV reservoir. Future studies could enlarge the sample size to verify our conclusion.

Our study has several limitations. First, we did not include healthy individuals to immunize HBV vaccine as controls; thus, we can only compare the changes in immune levels before and after vaccination. Second, this study only observed the changes in the number of cells secreting IFN-r and TNF-a before and after vaccination by the ELISPOT method. Multicolor flow cytometry has not observed changes in the proportion of CD4^+^ and CD8^+^ cell subsets and the level of cytokines before and after vaccination. Third, the enrolled PLWHs were not vaccinated in the last 5 years, but the proportion of those vaccinated before this period was unknown. Finally, the sample size was small, and we have not followed up with these patients for a long time. The existence of the limitations mentioned above made us more cautious when interpreting the research results. Therefore, we chose to simultaneously evaluate the humoral and cellular immune responses after HBV vaccination among PLWHs with successful treatment. Furthermore, we demonstrated the dynamics of the total HIV DNA levels in these patients, which were rarely shown in previous studies. We hope these measures make our results more scientific and comprehensive.

## 5. Conclusions

In conclusion, PLWH with successful treatment could achieve ideal humoral immune responses and persistent anti-HBs titers after the completion of a standard HBV vaccination schedule. Unfortunately, the cellular immune response was not optimal, and HBV vaccination did not have an effect on the total HIV DNA levels.

## Figures and Tables

**Figure 1 vaccines-11-00400-f001:**
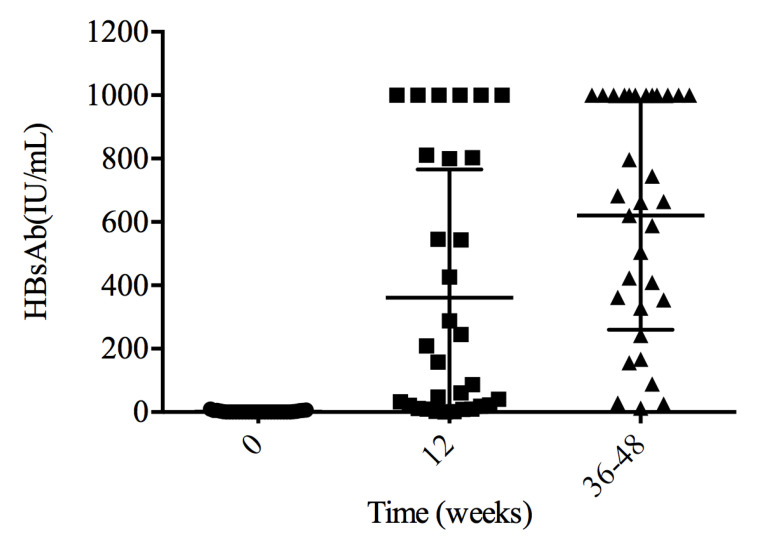
HBsAb levels after primary vaccination. All patients achieved a protective HBsAb titer between weeks 36 and 48.

**Figure 2 vaccines-11-00400-f002:**
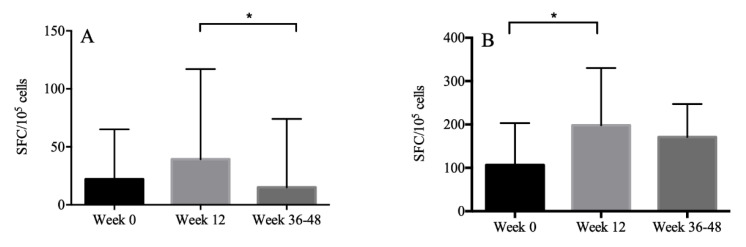
The magnitude of the IFN-r (**A**) and TNF-a (**B**) response to HBsAg presented with median and interquartile ranges. The level of IFN-r secreted at week 12 was higher than that at week 36, and the level of TNF-a secreted at week 12 was higher than that at week 0. *: *p* < 0.05.

**Figure 3 vaccines-11-00400-f003:**
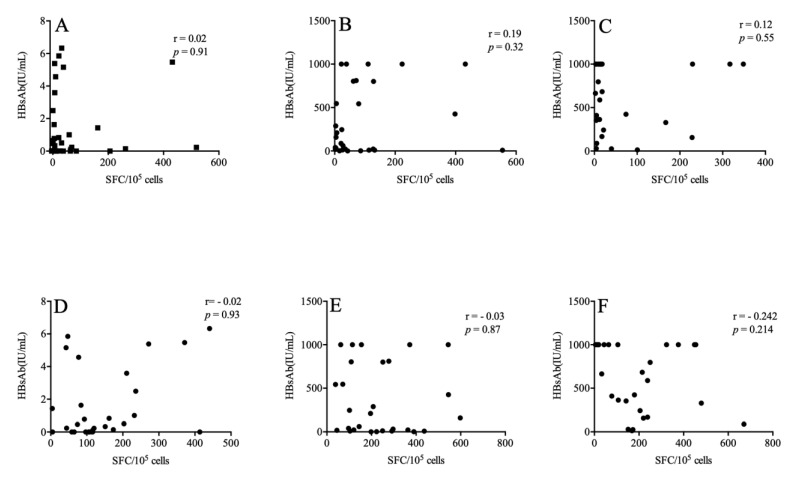
The association between HBsAg-specific T cell responses and the HBsAb titer. The above figures describe the relationships between HBsAb and IFN-r at weeks 0 (**A**), 12 (**B**) and 36–48 (**C**) respectively. The following are the relationship between HBsAb and TNF-a at weeks 0 (**D**), 12 (**E**) and 36–48 (**F**).

**Table 1 vaccines-11-00400-t001:** Pre-vaccine characteristics of enrolled patients.

Variables	N = 32
Sex, n (%)	
Men	29 (90.6)
Age (years, median (IQR))	36 (30–50)
Transmission route, n (%)	
Homosexual	24 (75.0)
Heterosexual	3 (9.38)
Bi-sexual	1 (3.13)
Blood	4 (12.5)
Weeks of cART initiation (median (IQR))	72 (44–93)
cART regimen, n (%)NRTIs+NNRTIs	23 (71.9)
NRTIs+PIs	4 (12.5)
NRTIs+INSTIs	4 (12.5)
NRTIs+PIs+INSTIs	1 (3.1)
HBsAb titer (IU/mL, median (IQR))	0.40 (0.0–2.28)
ALT (U/L, median (IQR))	25.5 (16.8–35.0)
AST (U/L, median (IQR))	24.0 (21.0–30.5)
TBil (µmol/L, median (IQR))	8.8 (6.8–11.9)
GGT (U/L, median (IQR))	32.5 (25.8–43.3)
ALP (U/L, median (IQR))	93 (76–107)
Cr(µmol/L, median (IQR))	82 (70.8–89.8)

NRTIs: nucleotide reverse transcriptase inhibitors, NNRTIs: non-nucleotide reverse transcriptase inhibitors, PIs: Protease inhibitors, INSTIs: integrase strand transfer inhibitors, ALT: Alanine aminotransferase, AST: Aspartate aminotransferase, TBil: total bilirubin, GGT: r-glutamyl transpeptidase, ALP: alkaline phosphatase, Cr: creatinine.

**Table 2 vaccines-11-00400-t002:** Characteristics of participants who did not sustain protective HBV serological immunity at 102 weeks.

	Patient 1	Patient 2	Patient 3
Age	35	42	58
Gender	Men	Men	Men
nadir CD4 count	273	383	236
type of ART regimen	3TC+TDF+EFV	3TC+TDF+RAL	3TC+TDF+EFV
Weeks of cART initiation (median (IQR))	272	176	360
HBsAb at week 12	22.04	209.18	18.13

**Table 3 vaccines-11-00400-t003:** Characteristics of participants who sustained protective HBV serological immunity at 102 weeks.

Variables	N = 29
Age (years, median (IQR))	35 (30–49)
Sex, n (%)	
Men	26
cART regimen, n (%)	
NRTIs+NNRTIs	21
NRTIs+PIs	4
NRTIs+INSTIs	3
NRTIs+PIs+INSTIs	1
Weeks of cART initiation (median (IQR))	288 (150–382)
HBsAb at week 12	201.5 (13.8–808.5)

## Data Availability

Datasets used in this analysis are available from the corresponding author on reasonable request.

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
