# Peer review of "Immune Responses to HBV Vaccine in People Living with HIV (PLWHs) Who Achieved Successful Treatment: A Prospective Cohort Study"

_vaccines, 2023, doi:10.3390/vaccines11020400_

Round 1

Reviewer 1 Report

Comments:

1)    Authors refer to IFN-r throughout. Are you measuring IFN-gamma specifically?

2)    A simple timeline showing vaccination and blood/measurement time points would be quite useful to the reader.

3)    More information on T cell phenotyping is needed. The methods does not discuss how naïve, central, and effector memory cells were identified and quantified.

4)    A little more information on Figure 2 would help. In the figure legend, please indicate what the error bars in the figure represent, as this will help the reader understand how significance is met with such large error bars. In addition, the level of IFN and TNF at different time points as listed in the text does not appear to match what the bars show in Figure 2. Can the authors please crosscheck this?

5)    What are the panels in Figure 3? While I understand that there is no correlation, it is unclear what is being shown in different panels.

6)    What specific HepB vaccine was used in this study? Were all patients given the same vaccine?

7)   The children in reference 16 received both HAV and HBV vaccines. This should be mentioned.

Author Response

                                                                                                   Date: Jan 14, 2023

Dear Editor-in-Chief:

Thank you very much for your comments and suggestions regarding our manuscript entitled, “Immune responses to HBV vaccine in persons living with HIV (PLWHs) who achieved successful treatment: a prospective cohort studyThe suggestions and comments are important and useful for the further improvement of our manuscript. The manuscript has been carefully rechecked, and appropriate changes have been made in accordance with your comments and suggestions. The responses to comments have been prepared and enclosed herewith. I would like to resubmit this revised manuscript to the Vaccines, and hope it is now suitable for publication.

With kindest regards,

Very respectfully,

Taisheng Li

Department of Infectious Diseases

Peking Union Medical College Hospital

Chinese Academy of Medical Sciences

Beijing, China 100730

E-mail:  litsh@263.net

The following document details our point-by-point responses to the reviewers and indicates the changes made in this manuscript:

Reviewer: 1

Comments to the Author

1)    Authors refer to IFN-r throughout. Are you measuring IFN-gamma specifically?

Answer:Thankyou for your comment very much.

Yes. We measuredIFN-gamma specifically in this study. PBMCs (1.0*105) were pulsed with 1 ug/mL recombinant Hepatitis B surface protein for 24 hours at 37 degree centigrade in 5% CO2 and then we tested the IFN-gammalevel using a standard Human IFN-gammaELISpot assay in triplicate wells. We described the measurement of IFN-r and TNF-a levels in lines 132-141.

2)    A simple timeline showing vaccination and blood/measurement time points would be quite useful to the reader.

Answer:Thankyou for your suggestion and this suggestion is very constructive. We added a timeline according to your suggestion and presented it in supplementary materials. The patients were followed-up every 12 weeks in our clinics, therefore, HBsAb were tested and full blood were collected at weeks 0, 4, and 36 in the original plan. However, some patients can’t come for follow-up at weeks 36 due to the outbreak of the COVID-19 epidemic. So they were followed-up between weeks 36 and weeks 48.

Besides, anti-HBs were tested annually in every patients, whether vaccinated or not, in order to evaluate the HBV infection in our clinics. Therefore, we could observe the dynamics of anti-HBs titer in longer follow-up duration.

We added the above information in method section in lines 102-109.

3)    More information on T cell phenotyping is needed. The methods does not discuss how naïve, central, and effector memory cells were identified and quantified.

Answer:Thankyou for your advice. We explained the lymphocyte subsets phenotyping in lines 120-130 in our revised manuscript.

2.3. Lymphocyte subsets phenotyping

  Immunophenotyping of peripherial blood lymphocytes was analyzed by three-color flow cytometry (Epics XL flow cytometry; Bechman Coulter, USA) using commercially available monoclonal antibodies as previously described[22]. The percentages and counts of the following lymphocyte subsets were measured, including CD3-CD16+CD56+NK cells, CD3-CD4+T cells, CD3-CD8+T cells, memory CD4+T cells, CD4+ CD45RA+CD62+naïve T cells, CD8+HLA-DR+T cells, CD8+CD38+T cells, and the CD4/CD8 ratio. The memory CD4+T cells were counted by difference between CD4+T cells and naïve CD4+T cells. Cell counts of lymphocyte subsets were calculated using a dual-platform method with white blood cell counts and lymphocyte differentials obtained from routine blood tests of the same specimen.

4)    A little more information on Figure 2 would help. In the figure legend, please indicate what the error bars in the figure represent, as this will help the reader understand how significance is met with such large error bars. In addition, the level of IFN and TNF at different time points as listed in the text does not appear to match what the bars show in Figure 2. Can the authors please crosscheck this?

Answer:Thankyou for your question. We must acknowledge that we have made a mistake. We presented the magnitude of the IFN-r and TNF-a response to HBsAg with mean and standard error in Figure 2. Therefore, it did not matchthe level of IFN-a and TNF-r at different time points as listed in the text, which was shown with median and interquartile ranges. We modified it in the revised manuscript in lines 218-225 and lines 227-230.

The median magnitude of the IFN-r response to HBsAg was 22.0 (IQR: 6.5-65.0), 39.3 (IQR: 13.8-117.0), 15.0 (IQR: 5.5-74.0) SFC/105 PBMC, respectively at weeks 0, 12, 36 to 48. And the median magnitude of the TNF-a response to HBsAg was 106.5 (IQR: 58.5-203.0), 198.0 (IQR: 106.5-330.0), 170.8 (IQR: 48.6-247.1) SFC/105 PBMC, respectively at weeks 0, 12, 36 to 48. The level of IFN-r secreted at weeks 12 was higher than that at weeks 36 to 48 (p=0.02) and the level of TNF-a secreted at weeks 12 was higher than that at weeks 0 (p<0.001) (Figure 2).

Figure 2 The magnitude of the IFN-r (A) and TNF-a (B) response to HBsAg, which was presented with median and interquartile ranges.The level of IFN-r secreted at weeks 12 was higher than that at weeks 36, and the level of TNF-a secreted at weeks 12 was higher than that at week 0.

5)    What are the panels in Figure 3? While I understand that there is no correlation, it is unclear what is being shown in different panels.

Answer:Thankyou very much for your questions. The above figures described the relationships between HBsAb and IFN-r at weeks 0 (A), 12 (B) and 36-48 (C) respectively. And the followings were the relationship between HBsAb and TNF-a at weeks 0 (D), 12 (E) and 36-48 (F) respectively. We described it in Figure 3 legend in lines 237-239.

We explored the association between the HBsAg-specific T cells responses and the HBsAb titer during the follow-up period. Unfortunately, no correlation was found between the magnitude of the IFN-r, TNF-a response to HBsAg and the HBsAb titer after the immunization procedure. We showed these results in Figure 3.

Figure 3 The association between the HBsAg-specific T cells responses and the HBsAb titer. The above figures described the relationships between HBsAb and IFN-r at weeks 0 (A), 12 (B) and 36-48 (C) respectively. And the followings were the relationship between HBsAb and TNF-a at weeks 0 (D), 12 (E) and 36-48 (F) respectively.

6)    What specific HepB vaccine was used in this study? Were all patients given the same vaccine?

Answer:Thankyou very much for your questions. Recombinant HBV DNA vaccine was used in this study. And all patients were given the same vaccine. We covered that in methods section in line 101.

7)   The children in reference 16 received both HAV and HBV vaccines. This should be mentioned.

Answer:Thankyou for thereminder! We explained the children in reference 16 received both HAV and HBV vaccines in lines 75-76.

Reviewer 2 Report

Dear authors, 

I have read your interesting work about the immune responses to the HBV vaccine in people with HIV. I have some comments about your paper.

Title

I suggest modifying persons with people.

General comment

There are several typos in the manuscript and figures (e.g. figure S1 “ 100 with HBsAg and HBsAb negetive”).

All abbreviations should be written entirely in the first appearance in the text (e.g. INF, TNF).

When a sentence starts with a number, it should be written in letters.

Introduction

“Approximately 10% of persons living with human immunodeficiency virus (PLWH) are coinfected with hepatitis B virus (HBV)”. However, it is not clear if it is the worldwide prevalence, Chinese prevalence, or german prevalence since you cited only a German study (ref. 1).

Please use people living with HIV instead of HIV-infected patients.

I would like to read how the vaccination could improve the quality of life of PLWH. For example, many of them, doing a TDF or TAF-based regimens are protected from HBV infection (doi:10.1097/QAD.0000000000000180). However, some treatments, such as the new long-acting CAB+RPV, could not protect from HBV, and people with HBV are not eligible for this treatment (10.3390/jpm12020188). In this context, vaccination is strongly suggested for a better treatment choice and, consequently, a better quality of life.

Methods

It is not a prospective trial. It is a prospective study.

It is not clear the rationale of this inclusion criterion “without a history of HBV vaccination in previous five years.”. Why have you chosen five years? Probably it would be better if they had not received the vaccination.

It should be better to specify which vaccine you used in your study.

It should be better to specify which test you used to measure anti-HBs antibodies. I also suggest providing the cut-off of this test.

Which test have you used to verify the normality of the distribution?

It needs to be clarified whether Figure S1, Figure S2, and Figure S3 should be in the supplemental material. If they are supplemental material, why are they in the text?

Results

What do you mean by blood transmission? Are they PWID or get infected during some medical procedures?

You reported a CD4/CD8 ratio = 0.32. Is it the median of the nadir ratio?

In table 1, specify for each row if it is a number with percentage or median with IQR.

In table 1, provide in the caption the explanation of each abbreviation you used in the table.

Author Response

                                                                                                         Date: Jan 14, 2023

Dear Editor-in-Chief:

Thank you very much for your comments and suggestions regarding our manuscript entitled, “Immune responses to HBV vaccine in persons living with HIV (PLWHs) who achieved successful treatment: a prospective cohort studyThe suggestions and comments are important and useful for the further improvement of our manuscript. The manuscript has been carefully rechecked, and appropriate changes have been made in accordance with your comments and suggestions. The responses to comments have been prepared and enclosed herewith. I would like to resubmit this revised manuscript to the Vaccines, and hope it is now suitable for publication.

With kindest regards,

Very respectfully,

Taisheng Li

Department of Infectious Diseases

Peking Union Medical College Hospital

Chinese Academy of Medical Sciences

Beijing, China 100730

E-mail:  litsh@263.net

The following document details our point-by-point responses to the reviewers and indicates the changes made in this manuscript:

Reviewer: 2

Comments and Suggestions for Authors

Dear authors, 

I have read your interesting work about the immune responses to the HBV vaccine in people with HIV. I have some comments about your paper.

Title

I suggest modifying persons with people.

Answer:Thank you for your suggestion. We modified persons with people in the title.

General comment

There are several typos in the manuscript and figures (e.g. figure S1 “ 100 with HBsAg and HBsAb negetive”).

All abbreviations should be written entirely in the first appearance in the text (e.g. INF, TNF).

When a sentence starts with a number, it should be written in letters.

Answer:Thank you very much for your kind reminder. We looked through our manuscript again and modified the typos. All abbreviations have been written entirely in the first appearance in the text. And when a sentence starts with a number, it has been written in letters.

Introduction

“Approximately 10% of persons living with human immunodeficiency virus (PLWH) are coinfected with hepatitis B virus (HBV)”. However, it is not clear if it is the worldwide prevalence, Chinese prevalence, or german prevalence since you cited only a German study (ref. 1).

Answer:Thank you for your question. We have modified this contents and provided worldwide HIV/HBV coinfection prevalence in introduction section in lines 41-44.  

Approximately 7.5% of people living with human immunodeficiency virus (PLWH)worldwide are coinfected with hepatitis B virus (HBV) due to the same transmission routes[1]. The coinfection prevalence varies in different regions. For example, it is higher than 10% in sub-Saharan Africa, west Africa, east Asia, and India[1, 2].

Reference

  1. Bollinger, R.C., et al., Addressing the global burden of hepatitis B virus while developing long-acting injectables for the prevention and treatment of HIV. Lancet HIV, 2020. 7(6): p. e443-e448.
  2. Singh, K.P., et al., HIV-hepatitis B virus coinfection: epidemiology, pathogenesis, and treatment. AIDS, 2017. 31(15): p. 2035-2052.

Please use people living with HIV instead of HIV-infected patients.

Answer:Thank you for your suggestion. We usedpeople living with HIV (PLWHs)instead of HIV-infected patients in our revised manuscript.

I would like to read how the vaccination could improve the quality of life of PLWH. For example, many of them, doing a TDF or TAF-based regimens are protected from HBV infection (doi:10.1097/QAD.0000000000000180). However, some treatments, such as the new long-acting CAB+RPV, could not protect from HBV, and people with HBV are not eligible for this treatment (10.3390/jpm12020188). In this context, vaccination is strongly suggested for a better treatment choice and, consequently, a better quality of life.

Answer:Thank you very much for your advice. We have added the importance of HBV vaccination in PLWHs according to your suggestion in lines 46-50.

Some patients treated with a TDF or TAF-based regimens are protected from HBV infection[5]. However, the new long-acting drugs such as cabotegravir and rilpivirine, could not protect from HBV, and people with HBV are not eligible for this treatment[6]. Therefore, vaccination is strongly suggested for a better treatment choice and, consequently, a better quality of life in PLWHswith negative hepatitis B serology[7].

  1. De Vito, A., et al., Could Long-Acting Cabotegravir-Rilpivirine Be the Future for All People Living with HIV? Response Based on Genotype Resistance Test from a Multicenter Italian Cohort.J Pers Med, 2022. 12(2).
  2. Schillie, S., et al., Prevention of Hepatitis B Virus Infection in the United States: Recommendations of the Advisory Committee on Immunization Practices.MMWR Recomm Rep, 2018. 67(1): p. 1-31.

Methods

It is not a prospective trial. It is a prospective study.

Answer:Thank you very much for your suggestion. We used a prospective study instead of a prospective trial in line 90.

It is not clear the rationale of this inclusion criterion “without a history of HBV vaccination in previous five years.”. Why have you chosen five years? Probably it would be better if they had not received the vaccination.

Answer:Thank you very much for your question. China has promoted hepatitis B vaccination nationwide since 1992. Therefore, most young adults have received HBV vaccine at birth. We were not able to enroll enough patients if we included people who had not received the vaccination. The effective time of HBV vaccines is five years in China. Therefore, we enrolled PLWH without a history of HBV vaccination in previous five years in order to eliminate the interference.

It should be better to specify which vaccine you used in your study.

Answer:Thank you very much for your suggestion. Recombinant HBV DNA vaccine was used in this study. We added it in our manuscript in lines 101.

It should be better to specify which test you used to measure anti-HBs antibodies. I also suggest providing the cut-off of this test.

Answer:Thank you very much for your advice! Enzyme linked immunosorbent assay (Elisa) was used to measure HBsAb and the cut-off value of this test 10 IU/mL. We added this information in lines 109-111 in revised manuscript.

Which test have you used to verify the normality of the distribution?

Answer:Thank you very much for your question. Shapiro-Wilk test was used to verify the normality of the distribution. p > 0.05 was considered as normality of the distribution.

It needs to be clarified whether Figure S1, Figure S2, and Figure S3 should be in the supplemental material. If they are supplemental material, why are they in the text?

Answer:Thank you very much for your advice! Figure S1, Figure S2, and Figure S3 are supplemental material and they are in the supplemental material now.

Results

What do you mean by blood transmission? Are they PWID or get infected during some medical procedures?

Answer:Thank you very much for your question. Blood transmission means getting infected during some medical procedures in our manuscript.

You reported a CD4/CD8 ratio = 0.32. Is it the median of the nadir ratio?

Answer:Thank you very much for your question. No, CD4/CD8 ratio was 0.32 (IQR: 0.19-0.38) before cART.

In table 1, specify for each row if it is a number with percentage or median with IQR.

Answer:Thank you very much for your suggestion. We have specified for each row when it was a number with percentage or median with IQR in table 1.

Table 1 Pre-vaccine characteristics of enrolled patients

Variables

N=32

Sex, n (%)

Men

29 (90.6)

Age (years, median (IQR))

36 (30-50)

Transmission route, n (%)

Sexual

28 (87.5)

Blood

4 (12.5)

Weeks of cART initiation (median (IQR))

72 (44-93)

cART regimen, n (%)

NRTIs+NNRTIs

23 (71.9)

NRTIs+PIs

4 (12.5)

NRTIs+INSTIs

4 (12.5)

NRTIs+PIs+INSTIs

1 (3.1)

HBsAb titer (IU/mL, median (IQR))

0.40 (0.0-2.28)

ALT (U/L, median (IQR))

25.5 (16.8-35.0)

AST (U/L, median (IQR))

24.0 (21.0-30.5)

TBil (µmol/L, median (IQR))

8.8 (6.8-11.9)

GGT (U/L, median (IQR))

32.5 (25.8-43.3)

ALP (U/L, median (IQR))

93 (76-107)

Cr(µmol/L, median (IQR))

82 (70.8-89.8)

In table 1, provide in the caption the explanation of each abbreviation you used in the table.

Answer:Thank you very much for your suggestion. We provided in the caption the explanation of each abbreviation we used in the table 1 in lines 167-171.

NRTIs: nucleotide reverse transcriptase inhibitors, NNRTIs: non-nucleotide reverse transcriptase inhibitors, PIs: Protease inhibitors, INSTIs: integrase strand transfer inhibitors, ALT: Alanine aminotransferase, AST: Aspartate aminotransferase, TBil: total bilirubin, GGT: r-glutamyl transpeptidase, ALP: alkaline phosphatase, Cr: creatinine

Reviewer 3 Report

This paper reports on the immunological response to HBV immunization among PLWH. This topic is of importance as it provides insights on the long-term immunological and cellular response to HBV immunization among successfully treated PLWH. In addition, it assessed the potential added value of HBV immunization on HIV reservoirs.

However, this article suffers from a number of limitations that restricts the interpretation of the findings.

Major comments:

-          Participants were not vaccinated in the last 5 years but the proportion of those vaccinated before this period would be informative. Some of them had low titers of HBsAb at baseline, suggesting some previous vaccination. This information is important to interpret the findings.

-          The participant follow-up is given as up to 36-48 weeks in the methods, while HBsAb titers are presented at 102 weeks, i.e. about 18 months post complete immunization. No data on the cellular response are presented at this follow-up visit. If the reason is because the cellular response did not mount after week 36-48 and was therefore very unlikely to be increased at 102 weeks, it would be useful to mention it in the methods.

-          A sample size or power calculation would also be useful to estimate the increase in cellular response that could be detected between week 12, 36-48 and baseline with 32 patients.

-          The representativeness of participants raises concern.  Among those eligible, only half were enroled, and 90% of them were males which is not the case among most PLWH infected through sexual transmission worldwide.

-          It would be interesting to describe further the 3 participants who did not sustain a protective HBV serological immunity at 102 weeks and those who did, in terms of age, gender, HIV data (nadir CD4 count, type of ART regimen, duration of ART, etc.) and HBV immune response at week 12.

-          Line 247: "we found that T-cell response to HBsAg 247 was higher at weeks 12 than at weeks 0 and 36". This is not supported by the data which don't seem to show a statistical difference between baseline and week 12. Therefore, the interpretation following this wrong finding (Lines 247-253) should be tempered.

-          The study limitations are well addressed, including their importance for the interpretation of their findings.

-          The final sentence of the paper is not addressed in the discussion and is not supported by the data as all participants receive the same immunization dosage.

More minor comments

Methods

-          The 102 follow-up visit and its planned content before the study should be added.

Results

-          The median follow-up time for the visit 36-48 weeks would be informative.

-          Table 1: please clarify whether values in the right column are median and IQR or means

-          3 patients had detectable HIV RNA during follow-up. Were two of them the patients with detectable HIV RNA at baseline?

-          Figure 2: the statistical comparison of the level of cytokine secretion would be more relevant using time 0 as reference (i.e. week 12 vs. baseline & week 36 vs. baseline). This result is recalled in the discussion section (line 274) but it should be stated in the results section as well.

-          Figure 3 results: were samples taken after the 1st immunization? Why this assessment was not made after the completion of the immunization scheme (i.e. 3rd dose)?

Author Response

                                                                                                     Date: Jan 14, 2023

Dear Editor-in-Chief:

Thank you very much for your comments and suggestions regarding our manuscript entitled, “Immune responses to HBV vaccine in persons living with HIV (PLWHs) who achieved successful treatment: a prospective cohort studyThe suggestions and comments are important and useful for the further improvement of our manuscript. The manuscript has been carefully rechecked, and appropriate changes have been made in accordance with your comments and suggestions. The responses to comments have been prepared and enclosed herewith. I would like to resubmit this revised manuscript to the Vaccines, and hope it is now suitable for publication.

With kindest regards,

Very respectfully,

Taisheng Li

Department of Infectious Diseases

Peking Union Medical College Hospital

Chinese Academy of Medical Sciences

Beijing, China 100730

E-mail:  litsh@263.net

The following document details our point-by-point responses to the reviewers and indicates the changes made in this manuscript:

Reviewer: 3

Comments and Suggestions for Authors

This paper reports on the immunological response to HBV immunization among PLWH. This topic is of importance as it provides insights on the long-term immunological and cellular response to HBV immunization among successfully treated PLWH. In addition, it assessed the potential added value of HBV immunization on HIV reservoirs. 

However, this article suffers from a number of limitations that restricts the interpretation of the findings. 

Major comments: 

- Participants were not vaccinated in the last 5 years but the proportion of those vaccinated before this period would be informative. Some of them had low titers of HBsAb at baseline, suggesting some previous vaccination. This information is important to interpret the findings. 

Answer:Thank you very much for your suggestion. The proportion of patients vaccinated before was important to interpret the findings. We tried to ask the enrolled patients this question. Unfortunately, some of them could not answer this question exactly because it’s been a long time. We had to acknowledge it was one of the limitations and we described it in the limitations in line 317-318.

- The participant follow-up is given as up to 36-48 weeks in the methods, while HBsAb titers are presented at 102 weeks, i.e. about 18 months post complete immunization. No data on the cellular response are presented at this follow-up visit. If the reason is because the cellular response did not mount after week 36-48 and was therefore very unlikely to be increased at 102 weeks, it would be useful to mention it in the methods. 

Answer:Thank you very much for your advice! Yes, the cellular responses were presented as up to 36-48 weeks because the cellular responses did not mount after week 36-48 and was therefore very unlikely to be increased at 102 weeks. Therefore, we did not observe the dynamics of the cellular response after weeks 48. And HBsAb were tested annually in every patients, whether vaccinated or not, in order to evaluate the HBV infection in our clinics. So we could observe the dynamics of HBsAb titer in longer follow-up duration. We added this information in the methods in lines 102-109.

- A sample size or power calculation would also be useful to estimate the increase in cellular response that could be detected between week 12, 36-48 and baseline with 32 patients. 

Answer:Thank you very much for your suggestion. We felt terribly sorry that we had no idea how to calculate the sample size. We tried to study the relevant knowledge of sample size calculation for single group design. But the sample size was calculated by the average value with standard error. Our data did not obey the normal distribution, so we had to use the median with interquartile ranges to analyze the data, but we did not find how to use the median with interquartile ranges to calculate the sample size. We would like to ask the reviewer how to calculate the sample size in this case. We really want to learn and master it.

- The representativeness of participants raises concern.  Among those eligible, only half were enrolled, and 90% of them were males which is not the case among most PLWH infected through sexual transmission worldwide. 

Answer:Thank you very much for your question. 90% of the enrolled participants were men and most of them were infected by homosexual transmission in our study. These results obtaining from the single center were more representative in MSM. Larger multi-center researches could be carried out to promote these results in the future.

- It would be interesting to describe further the 3 participants who did not sustain a protective HBV serological immunity at 102 weeks and those who did, in terms of age, gender, HIV data (nadir CD4 count, type of ART regimen, duration of ART, etc.) and HBV immune response at week 12. 

Answer:Thank you very much for your advice! We described further the 3 participants who did not sustain a protective HBV serological immunity at 102 weeks and those who did. Only 3 patients did not sustain a protective HBV serological immunity at 102 weeks, which made difficult to describe it with median and IQR. Therefore, we described the characteristics of the two groups patients in Table 2 and Table 3, respectively. We added Table 2 and Table 3 in lines 207-214.

Table 2 Characteristics of participants who did not sustain a protective HBV    serological immunity at 102 weeks

Patient 1

Patient 2

Patient 3

Age

35

42

58

Gender

Men

Men

Men

nadir CD4 count

273

383

236

type of ART regimen

3TC+TDF+EFV

3TC+TDF+RAL

3TC+TDF+EFV

Weeks of cART initiation (median(IQR))

272

176

360

HBsAb at weeks 12

22.04

209.18

18.13

Table 3 Characteristics of participants who sustained a protective HBV   serological immunity at 102 weeks

Variables

N=29

Age (years, median (IQR))

35(30-49)

Sex, n (%)

Men

26

cART regimen, n (%)

NRTIs+NNRTIs

21

NRTIs+PIs

4

NRTIs+INSTIs

3

NRTIs+PIs+INSTIs

1

Weeks of cART initiation (median (IQR))

288(150-382)

HBsAb at weeks 12

201.5(13.8-808.5)

- Line 247: "we found that T-cell response to HBsAg 247 was higher at weeks 12 than at weeks 0 and 36". This is not supported by the data which don't seem to show a statistical difference between baseline and week 12. Therefore, the interpretation following this wrong finding (Lines 247-253) should be tempered. 

Answer:Thank you very much for your advice. Our results showed that the level of TNF-a secreted at weeks 12 was higher than that at week 0 while the level of IFN-r secreted at weeks 12 was comparable with that at week 0. The interpretation following these results seemed not suitable. Therefore, we deleted “We thought this may be the result of repeat stimulation in a short time. Patients immunized with the third HBV vaccine five months after the second vaccine. The long duration may decrease the magnitude of T-cell response to HBsAg. These results indicated that receiving vaccinations on a rapidly accelerated schedule may be helpful to promote cellular immune responses. Studies in the future may warrant consideration to immunized HIV-infected patients to elevate IFN-r and TNF-a levels.” in the revised manuscript.

- The study limitations are well addressed, including their importance for the interpretation of their findings. 

Answer:Thank you very much for your suggestion. We added the impact of study limitations on the interpretation of our findings in lines 319-325.

The existence of limitations above made us more cautious when interpreting the research results. Therefore, we chose to evaluate the humoral and cellular immune responsessimultaneously after HBV vaccination among PLWHs with successful treatment. What’s more, we demonstrated the dynamics of the total HIV DNA levels in these patients, which was rarely shown in previous studies. We hope these measures make our results more scientific and comprehensive.

-The final sentence of the paper is not addressed in the discussion and is not supported by the data as all participants receive the same immunization dosage. 

Answer:Thank you very much for your suggestion. We deleted the sentence “Standardized or double-dose regimens at monthly intervals warrant consideration in future studies.” in the revised manuscript.

More minor comments

Methods

-  The 102 follow-up visit and its planned content before the study should be added. 

Answer:Thank you very much for your advice. The patients were followed-up every 12 weeks in our clinics, therefore, anti-HBs were tested and full blood were collected at weeks 0, 4, and 36 in the original plan. However, some patients can’t come for follow-up at weeks 36 due to the outbreak of the COVID-19 epidemic. So they were followed-up between weeks 36-48.

Besides, anti-HBs were tested annually in every patients, whether vaccinated or not, in order to evaluate the HBV infection in our clinics. Therefore, we could observe the dynamics of anti-HBs titer in longer follow-up duration.

Results

- The median follow-up time for the visit 36-48 weeks would be informative. 

Answer:Thank you very much for your suggestion. 20 patients were followed-up at weeks 36. 10 patients were followed-up at weeks 48. And the rest two of these patients were followed up at weeks 40 and 44. The median follow-up time was 36 (IQR: 36-48) weeks.

- Table 1: please clarify whether values in the right column are median and IQR or means

Answer:Thank you very much for your advice! The values in the right column are number with percentage or median and IQR and we have specified for each values in the table 1.

Table 1 Pre-vaccine characteristics of enrolled patients

Variables

N=32

Sex, n (%)

Men

29 (90.6)

Age (years, median (IQR))

36 (30-50)

Transmission route, n (%)

Sexual

28 (87.5)

Blood

4 (12.5)

Weeks of cART initiation (median (IQR))

72 (44-93)

cART regimen, n (%)

NRTIs+NNRTIs

23 (71.9)

NRTIs+PIs

4 (12.5)

NRTIs+INSTIs

4 (12.5)

NRTIs+PIs+INSTIs

1 (3.1)

HBsAb titer (IU/mL, median (IQR))

0.40 (0.0-2.28)

ALT (U/L, median (IQR))

25.5 (16.8-35.0)

AST (U/L, median (IQR))

24.0 (21.0-30.5)

TBil (µmol/L, median (IQR))

8.8 (6.8-11.9)

GGT (U/L, median (IQR))

32.5 (25.8-43.3)

ALP (U/L, median (IQR))

93 (76-107)

Cr(µmol/L, median (IQR))

82 (70.8-89.8)

- 3 patients had detectable HIV RNA during follow-up. Were two of them the patients with detectable HIV RNA at baseline? 

Answer:Thank you very much for your question. Yes, two of the patients had detectable HIV RNA, 79 copies/mL, 68 copies/mL, respectively at the first dose of their immunization (week 0). We added this information in lines 181-182 in revised manuscript.

- Figure 2: the statistical comparison of the level of cytokine secretion would be more relevant using time 0 as reference (i.e. week 12 vs. baseline & week 36 vs. baseline). This result is recalled in the discussion section (line 274) but it should be stated in the results section as well. 

Answer:Thank you very much for your suggestion. We compared the level of cytokine secretion at weeks 12 and 36 using week 0 as reference and stated it in the discussion section (lines 294-296).

- Figure 3 results: were samples taken after the 1st immunization? Why this assessment was not made after the completion of the immunization scheme (i.e. 3rd dose)?

Answer:Thank you very much for your question. We feel terribly sorry that we did not explain it clearly. The above figures describe the relationships between HBsAb and IFN-r at weeks 0 (A), 12 (B) and 36-48 (C) respectively. And the followings describe the relationship between HBsAb and TNF-a at weeks 0 (D), 12 (E) and 36-48 (F) respectively. We described it in figure legend in lines 237-239.

Round 2

Reviewer 2 Report

Thank you for replying to all my comments. I have re-read the manuscript and in my opinion is now suitable for publication.

Author Response

Thank you very much for your comments!

Reviewer 3 Report

Overall, the authors have addressed the comments appropriately. Few  comments though:

 - The authors should mention that 90% of their sample were MSM (or that MSM were clients of the recruitment site)

 - Only 50% of eligible patients participated. The main reason (refusal) should be incldued in the text. Reading again Figure S1, it is unusual that 11 participants refused to be vaccinated just after they signed an informed consent which stated that they are proposed to participate in a HBV vaccine study and that they will be vaccinated if they agree to participate. Do the authors have any tentative explanation for this situation ? 

The sample size calculation for the cellular response is not crucial. Of note, the distribution to look at is the difference between values at say week 12 and baseline, not the baseline values alone. If this distribution is not normal, you could have either tried and normalized the data by using some transformation (ex: log, 1/x, etc..) or use a non parametric test, but I agree the sample size calculation is less easy to interpret in this case.

Author Response

The following document details our point-by-point responses to the reviewer and indicates the changes made in this manuscript:

Reviewer: 3

Comments and Suggestions for Authors

Overall, the authors have addressed the comments appropriately. Few  comments though: 

 - The authors should mention that 90% of their sample were MSM (or that MSM were clients of the recruitment site)

Answer:Thankyou very much for your suggestion. We described the transmission route according to your suggestion in results section (lines 146-147) and Table 1 in the revised manuscript.

75.0% of individuals were infected with HIV by homosexual transmission route.

Table 1. Pre-vaccine characteristics of enrolled patients.

Variables

N=32

Sex, n (%)

Men

29 (90.6)

Age (years, median (IQR))

36 (30-50)

Transmission route, n (%)

Homosexual

Heterosexual

Bi-sexual

24(75.0)

3(9.38)

1(3.13)

Blood

4 (12.5)

Weeks of cART initiation (median (IQR))

72 (44-93)

cART regimen, n (%)

NRTIs+NNRTIs

23 (71.9)

NRTIs+PIs

4 (12.5)

NRTIs+INSTIs

4 (12.5)

NRTIs+PIs+INSTIs

1 (3.1)

HBsAb titer (IU/mL, median (IQR))

0.40 (0.0-2.28)

ALT (U/L, median (IQR))

25.5 (16.8-35.0)

AST (U/L, median (IQR))

24.0 (21.0-30.5)

TBil (µmol/L, median (IQR))

8.8 (6.8-11.9)

GGT (U/L, median (IQR))

32.5 (25.8-43.3)

ALP (U/L, median (IQR))

93 (76-107)

Cr(µmol/L, median (IQR))

82 (70.8-89.8)

 - Only 50% of eligible patients participated. The main reason (refusal) should be included in the text. Reading again Figure S1, it is unusual that 11 participants refused to be vaccinated just after they signed an informed consent which stated that they are proposed to participate in a HBV vaccine study and that they will be vaccinated if they agree to participate. Do the authors have any tentative explanation for this situation ? 

Answer: Thank you very much for your question. We found that we have made a mistake. It was 10 participants who did not vaccinate for personal reasons. We modified it in the revised Figure S1.

The main reason was that they need to take three leave to vaccinate, which they thought would affect their work. Furthermore, they still had concerns about the safety and effectiveness of vaccines.

We added the above information in lines 92-95 in the revised manuscript.

The sample size calculation for the cellular response is not crucial. Of note, the distribution to look at is the difference between values at say week 12 and baseline, not the baseline values alone. If this distribution is not normal, you could have either tried and normalized the data by using some transformation (ex: log, 1/x, etc..) or use a non parametric test, but I agree the sample size calculation is less easy to interpret in this case.

Answer:Thankyou very much for your comment.